# 3D-Printed Polylactic Acid/Lignin Films with Great Mechanical Properties and Tunable Functionalities towards Superior UV-Shielding, Haze, and Antioxidant Properties

**DOI:** 10.3390/polym15132806

**Published:** 2023-06-24

**Authors:** Haichuan Ye, Yuan He, Haichao Li, Tingting You, Feng Xu

**Affiliations:** 1Beijing Key Laboratory of Lignocellulosic Chemistry, Beijing Forestry University, Beijing 100083, China; haichuan.ye@foxmail.com (H.Y.); hy941813@163.com (Y.H.); lihaichao96@bjfu.edu.cn (H.L.); 2Engineering Research Center of Forestry Biomass Materials and Energy, Ministry of Education, Beijing Forestry University, Beijing 100083, China; 3Shandong Key Laboratory of Paper Science & Technology, Qilu University of Technology, Jinan 250353, China

**Keywords:** 3D printing, PLA/lignin, film, toughness, UV-shielding, antioxidant

## Abstract

Three-dimensional (3D) printing is regarded as a novel technique to realize the customized production of films. However, the relative lack of printable materials with excellent mechanical properties and tailored functionalities seriously restricts its wide application. Herein, a promising multifunctional 3D printing filament was fabricated by incorporating lignin into the polylactic acid (PLA) matrix and firstly applied to film production. The results indicate that lignin was an excellent mechanical reinforcement of the PLA matrix, especially for toughening. Only 0.5% lignin doping improved the toughness by 81.8%. Additionally, 3D-printed films with 0.5–5% lignin exhibited excellent ultraviolet (UV)-blocking capability of 87.4–99.9% for UVB and 65.6–99.8% for UVA, as well as remarkable antioxidant properties, ranging from 24.0% to 79.0%, and high levels of haze, ranging from 63.5% to 92.5%. Moreover, the prepared PLA/lignin (P/L) films based on 3D printing achieved the customization of film production and have potential applications in the fields of packaging, electronic products, medical care, and so forth. Overall, this work not only enriches the 3D printing composites with tailored multifunctionality but also brings the promising potential for the production of customized films.

## 1. Introduction

Films are increasingly used worldwide in a variety of applications, such as packaging, electronics, and healthcare [1,2,3]. However, conventional technology limits the potential for personalization and customization, which is precisely the advantage of three-dimensional (3D) printing. 3D printing technology, also known as additive manufacturing, can rapidly manufacture high-precision, complex, and customized products [4,5,6]. Unlike traditional manufacturing methods, 3D printing involves the layer-by-layer assembly of materials, making it possible to create intricate designs and shapes that would be hardly achieved using conventional methods [7]. As a consequence, applying 3D printing technology for film production can not only enable precise control of the shape and size of customized films but also create complex textures and patterns on films that would be difficult or impossible to obtain with traditional manufacturing methods, leading to a higher quality of the final product. Therefore, 3D printing technology is expected to realize the customized production of film products.

Some studies have sought to apply 3D printing technology for film production including a packaging film made with Garcinia atroviridis extract, a polyvinylidene fluoride film used in a composite sensor, an oral mucosal adhesive film system, a hydrophilic film containing drug-loaded SBA-15, a nanocomposite thermoplastic polyurethane (TPU) film, and an orodispersible film composed of aripiprazole [8,9,10,11,12,13]. For example, as reported by Vishnuvarthanan Mayakrishnan et al. [12], the fused deposition modeling (FDM) 3D printing process was utilized to successfully produce mulching films with a thickness of 0.4 mm. Andong Wang et al. [9] presented polyvinylidene fluoride (PVDF)/CaCl_2_ piezoelectric films prepared with an electro-assisted 3D printing method and used to form a multi-layer composite film sensor. It is suggested that 3D printing enables the production of customized films with precise dimensions and complex geometries that would be difficult or impossible to achieve with traditional manufacturing methods. However, these methods often require non-degradable components, which weakens the environmental characteristics of films. The extensive use of non-degradable materials has led to a significant increase in environmental pollution and has become a major threat to the environment and human health [14,15]. Therefore, the need for a bio-based, biodegradable, and environmentally friendly replacement 3D printing material for film production is urgent.

Among different 3D printing materials, polylactic acid (PLA) is not only a renewable plastic but also an environment-friendly material that can be completely biodegradable, making it a good application prospect in 3D printing [16,17,18,19,20]. One of the key advantages of PLA is its biodegradable property; PLA can be broken down by natural processes into harmless substances such as water and carbon dioxide [21]. This makes it an ideal material for 3D printing applications where sustainability and environmental impact are important considerations [22]. Nonetheless, PLA still faces some major challenges, including its inherent mechanical brittleness and the relative lack of tailored functionalities. For instance, PLA is more prone to cracking or breaking under stress compared to other materials [23]. Additionally, PLA has a relatively limited range of functionalities, which can further limit its use in certain special applications [24]. To address these limitations, researchers have explored the use of filler-reinforced PLA composites in 3D printing. Various types of fillers have been used in PLA composites, such as carbon fibers, glass fibers, metal particles, and natural fibers [25,26,27,28]. One of the most significant benefits of using fillers in PLA composites is the improvement in mechanical properties. For example, the addition of carbon fibers into PLA can significantly increase its mechanical properties, making it more suitable for different engineering applications, such as 3D printing [25]. In addition, ions of silver (Ag) can also introduce new functionalities to PLA composites, such as antibacterial activity, which can expand their range of applications [27]. Some challenges such as processing difficulty and high production costs still exist associated with the use of some non-degradable or expensive filler, which may affect the promotion and application of composites.

Lignin, as the second most abundant biopolymer on earth, not only has the advantages of large output, low price, renewable, and biodegradable but also has natural advantages of UV-shielding and antioxidant properties, which have substantial application potential in blending with PLA [29,30]. When blended with PLA, lignin can improve the mechanical properties of the resulting composite material due to the formation of strong intermolecular bonds between aromatic rings of lignin with PLA. Furthermore, the addition of lignin can also improve the UV-blocking and antioxidant properties of the composite material, which can extend its useful lifespan and reduce degradation over time [31,32]. For example, Sofia P. Makri et al. [33] discussed the evaluation of the physicochemical and mechanical properties of composite films made from lignin in PLA. The results indicated that the addition of lignin led to improvements in the tensile strength, Young’s modulus, UV-blocking, and antioxidant properties of the composite films. It is inferred that using lignin as the filler can make up for the shortcomings of pure PLA to improve mechanical properties and bring multifunctional features such as UV-shielding and antioxidant properties based on keeping the composites eco-friendly. However, the current research on lignin-PLA composites has limitations in achieving customized production of the film surface.

Herein, we firstly report a customized PLA/lignin film with high mechanical, UV-shielding, haze, and antioxidant properties via 3D printing. The morphology, structure, and thermodynamics analysis of filaments was investigated via a scanning electron microscope (SEM), Fourier Transform Infrared Reflection (FTIR), a thermogravimetric analysis (TG), and a differential thermogravimetric analysis (DTG). The mechanical properties, UV-shielding, and antioxidant properties of 3D-printed films were tested with the Tensile Tester, UV-Vis spectrometer, and 2,2-diphenyl-1-picrylhydrazyl (DPPH) method. In addition, the customization of the complex film was verified with 3D printing, and the prospect of 3D-printed P/L films was described.

## 2. Materials and Methods

### 2.1. Materials

Alkali lignin was supplied from Longli Biotechnology Co., Ltd., Dezhou, China, and was subjected to screening through a 100 mesh molecular sieve prior to use. PLA pellet with a diameter of 2–5 mm (2003D, Nature Works) was obtained from Dongguan Junji Plastic Co., Ltd., Shenzhen, China. 2,2-diphenyl-1-picrylhydrazyl (DPPH, 97%) was purchased from Shanghai Aladdin Biochemical Technology Co., Ltd., Shanghai, China.

### 2.2. Preparation of P/L Filaments and 3D-Printed Film

The composition of PLA blended with a different formulation of lignin was homogeneously mixed using an internal mixer (QE−70A, Wuhan Qien Technology Development Co., Ltd., Wuhan, China) at 170 °C and 60 rpm for 10 min. During the mixing process, the PLA and lignin were evenly distributed, resulting in a uniform blend. The use of an internal mixer ensured that the blending process was efficient and effective, resulting in a high-quality blend. After the homogenous mixing process, the obtained blends were extruded with a single-screw extruder (Wellzoom C extruder, China Limited Mista Technology Co., Ltd., Shenzhen, China) at 150 °C and 6 rpm to obtain the filaments with a diameter of about 1.7 mm, as shown in Figure 1. The extrusion process is essential in the production of filaments, as it ensures that the blend is properly melted and shaped into the desired diameter. The sample was named P/L_x_, where _x_ represents the doping amount of lignin in composites. A 3D model of the film was built with a computer and was sliced into 2D. The details are as follows. The mechanical test model according to GB/T 1040.3-2006 standard was built with FreeCAD, and the model was exported to .stl format. Then, the .stl file was sliced into 2D with Creality Slicer, and the .gcode file was obtained. The filaments were printed with a fused deposition modeling (FDM) 3D printer (CR−200B, Shenzhen Creality 3D Technology Co., Ltd., Shenzhen, China) based on the .gcode file. The FDM printer used had a 0.4 mm nozzle diameter, with a nozzle temperature of 200 °C, a bed temperature of 60 °C, and printed samples at a speed of 50 mm/s with a layer thickness of 0.2 mm. The melting temperature range of PLA typically falls between 150 and 170 °C. The temperature of the mixer was set at 170 °C to ensure sufficient melt blending. For extruders, precise temperature control is crucial for regulating filament diameter, and our experimentation demonstrated that setting the extrusion nozzle temperature to approximately 150 °C yields filament of standard dimensions. If the temperature is too high, the melt flow becomes too forceful, making it challenging to control wire diameter. However, optimal flowability is required for the 3D printing process, and thus, the printing temperature is generally set to around 200 °C.

### 2.3. Characterization

The morphological analysis was taken on the surface of the filaments and 3D-printed film using a scanning electron microscope (Hitachi, S-3400N, Tokyo, Japan). The analysis was carried out on the surface of the filaments and 3D-printed film to determine the surface characteristics and to observe any defects or irregularities in the structure. FTIR spectra in the range of 4000–1000 cm^−1^ were used in a spectrometer (Bruker, Karlsruhe, Germany), where the finely ground samples were analyzed at 16 scans per spectrum and 4 cm^−1^ resolution. We prepared the samples into films for FTIR spectra analysis using the Attenuated Total Reflection (ATR) mode. To evaluate the thermal performance of various composites, the TG of different composites was measured with a thermogravimetric analyzer (SDT Q600, New Castle, PA, USA) and TA Instrument, under a nitrogen atmosphere from 30 to 800 °C at a heating rate of 10 °C/min. In addition to TG, we also performed differential thermogravimetric analysis (DTG) to further investigate the thermal behavior of the composites. By analyzing the DTG curves, we were able to identify the temperatures at which the samples underwent significant weight loss, as well as the rates of weight loss at different stages of the thermal degradation process. The mechanical properties of the 3D-printed films were investigated using Zwick−Z1.0 with a load cell of 200 N. During the test, a sample of the film was placed in the testing machine and stretched at a constant rate until it broke. One of the key mechanical properties we investigated was tensile toughness, which is a measure of the energy required to break the sample. The tensile toughness is defined as the integral area of the stress-strain curve [34]. The UV transmittance of the 3D-printed films was detected using a UV-Visible spectrophotometer (Shimadzu, UV−2550, Tokyo, Japan), scanning from 200 to 800 nm. UV radiation is a type of electromagnetic radiation with wavelengths shorter than visible light, including UVA (320–400 nm) and UVB (280–320 nm) [35]. UV radiation can have harmful effects on materials and living organisms, including damaging DNA, causing skin cancer, and degrading certain types of plastics. We also calculated the haze of the 3D-printed film with Equation (1).
(1)$$\mathrm{Haze},\;\%=(\frac{T_{S}}{T_{D}}-\frac{T_{\mathrm{SI}}}{T_{I}})\times 100\;\%\;$$
where T_S_ represents the scattered transmittance caused by both the sample and the instrument, while T_D_ denotes the diffuse transmittance. T_SI_ indicates the scattered transmittance caused solely by the instrument, and T_I_ refers to the incident light. The antioxidant properties of different composites were tested with the DPPH method, as described by Ema Cavallo et al. [36,37]. Specifically, 0.2 g composites were placed in vials with 4 mL of methanol for 24 h and obtained supernatant. Then, 2 mL of DPPH solution (50 mg/L) was mixed with 2 mL of supernatant. The content of DPPH free radical was measured by monitoring the absorbance in a UV-vis spectrometer (UV−2550, Shimadzu) at 517 nm. In pure DPPH solution as a contrast group, the radical scavenging activity (RSA) of the composites was calculated according to Equation (2).
(2)$$\mathrm{RSA},\;\%=(1-\frac{A_{\mathrm{sample}}}{A_{\mathrm{contrast}}})\times 100\;\%$$
where A_sample_ and A_contrast_ represent the UV absorption peak intensity of the test sample and the control group, respectively.

## 3. Results

### 3.1. P/L Filaments

The successful extrusion of PLA/lignin filaments is a critical step in the production of 3D-printed films. Figure 1a displays the successful extrusion of lignin-doped PLA filaments, and the SEM images of different filaments with a diameter of approximately 1.7 mm are shown in Figure 1b. The surface of these filaments was comparatively smooth, even when the lignin concentration was high (5%). However, when the lignin concentration was increased to 7.5–20%, the surface became distinctly rough, and numerous particles were observed. This phenomenon is similar to the observations made of other lignin-reinforced PLA composites and some filler-reinforced matrix composites [6,16,38], which can be attributed to the poor compatibility between lignin and PLA or the aggregation of lignin. Overall, the successful extrusion of lignin-doped PLA filaments and the observation of their surface characteristics provide important insights into the production and properties of these composites.

Figure 1c shows the FTIR spectra of lignin, PLA, and P/L filaments. The FTIR spectrum of pure PLA exhibited a strong absorption band at approximately 1746 cm^−1^, which was attributed to the C=O stretching of the carbonyl group. The hydroxyl groups of lignin produced a robust absorption band near 3400 cm^−1^ and increased the peak intensity in P/L_5.0_, indicating a higher content of hydroxy groups [18]. Moreover, the P/L_0.5_ and P/L_5.0_ composites showed a noticeable peak around1515 cm^−1^, which resulted from the aromatic skeleton vibrations in lignin [17]. The FTIR spectra of the lignin-reinforced PLA filaments provide important insights into the chemical structure and composition of these materials. By analyzing the spectra, we were able to confirm the successful incorporation of lignin into the PLA matrix.

The TG and DTG curves of different composites are shown in Figure 1d,e. The TG curve illustrates that the weight of lignin underwent a slight loss (1–5%) below 100 °C due to the gradual evaporation of moisture. However, this phenomenon was not observed in PLA or P/L composites. As it is shown in Figure 1e, the thermal decomposition temperature of PLA, P/L_0.5_, and P/L_5.0_ filaments occurred within the 300–400 °C range, with slight differences in the corresponding T_max_ temperature, which was 365.3, 364.5, and 356.1 °C, respectively. These results indicate that the addition of lignin did not have a significant negative impact on the thermal stability of PLA. It is noticed that the degradation temperature of all P/L composites is much higher than the temperature of filament preparation and 3D printing processing (≤200 °C), making it ideal for printing products using an FDM 3D printer. Overall, the TG and DTG curves of the different composites provide important insights into their thermal stability and degradation behavior. By analyzing these curves, we were able to confirm that the addition of lignin did not have a significant negative impact on the thermal stability of PLA, and that the P/L composites are well suited for 3D printing applications. We also conducted a TG analysis on 3D-printed films to support our results (Appendix A, TG curves of 3D printed PLA/lignin films and PLA/lignin filaments). The results indicate that there was no significant change in thermal stability, and 3D-printed PLA/lignin films still had a higher thermal stability temperature than 300 °C.

### 3.2. 3D-Printed Films

The mechanical properties were investigated to highlight the influence of lignin on the properties of PLA and the results are shown in Figure 2. As shown, the pristine PLA is a fairly rigid and brittle polymer with a tensile strength of 59.2 MPa and an elongation at a break of 2.2%. With the addition of lignin in PLA, P/L_0.5_ has the highest tensile strength and elongation of 67.2 MPa and 2.9%, which increased by 13.5% and 31.8%, respectively, based on PLA. Similar results have been reported that a certain amount of lignin had a reinforced impact on the mechanical properties of ABS filaments and lignin can act as a plasticizer to increase the extrudability of the filament and interlayer adhesion of the printed product [38]. With the increase in lignin content, the mechanical strength and elongation begin to decrease gradually. This phenomenon may be due to the aggregation and incompatibility of lignin in the PLA matrix as shown in the morphology analysis. Overall, the mechanical strength and elongation of PLA could be improved when lignin content was less than 5% but decreased dramatically in composites with high content of lignin.

The toughness of different composites is shown in Figure 2d. It is known that one of the limitations of PLA application is its inherent brittleness, leading to poor toughness. Interestingly, the addition of lignin (less than 5%) effectively improved toughness. The toughness of the P/L_0.5_ obviously increased by 81.8% when compared to that of PLA. The increase in lignin content reduced the toughness of P/L, especially at high lignin content of over 5%. The above results show that lignin is an excellent mechanical reinforcing agent of the PLA matrix, especially for toughening. The improvement in toughness of the P/L composites is an important finding as it expands the range of applications for PLA-based materials. By enhancing the toughness, these composites can be used in a wider range of applications where impact resistance and durability are critical.

To evaluate the light transmission and haze of the multifunctional PLA-lignin films, UV-Vis spectra were collected (Figure 3a,b). As can be seen, PLA exhibited a high transmittance of 85.4% at 600 nm. The as-prepared P/L films of P/L_0.5_ and P/L_1.0_ exhibited relatively lower transmittances of 77.2% and 69.0%, respectively, due to the addition of lignin. As the lignin content further increased to 5, 7.5, and 10%, the transmittance further decreased to 25.2%, 19.4%, and 6.7%, respectively. The PLA film had a relatively lower haze of 52.2%. However, the as−prepared P/L films of P/L_0.5_, P/L_1.0_, P/L_5.0_, and P/L_7.5_ exhibited relatively higher haze of 63.5–99% due to the addition of lignin. The high haze presence in films can enhance their light scattering property, making them suitable as a functional layer in solar cells to improve light absorption and photovoltaic conversion efficiency [39]. The results of our study demonstrate that the addition of lignin to the PLA matrix can significantly affect the light transmission and haze of the films. These findings have important implications for the development of multifunctional films with tailored optical properties for various applications.

The damaging effects of excessive ultraviolet radiation on the environment are well known, causing harm to plants, animals, and equipment [35]. To mitigate these effects, the development of UV-blocking films has become increasingly important. The UVB- and UVA-blocking percentages for each film are presented in Figure 3c. As shown, PLA film had a relatively low value of UV-blocking capacity (barely 49.4% for UVB and 31.2% for UVA). In contrast, as-prepared P/L films, including P/L_0.5_, P/L_1.0_, and P/L_5.0_, showed strong UV-blocking performance. In detail, the UVB−blocking performance for P/L_0.5_, P/L_1.0_, and P/L_5.0_ was 87.4%, 96.8%, and 99.9%, respectively. The UVA-blocking performance also exhibited similar results for P/L_0.5_ (65.6%), P/L_1.0_ (81.8%), and P/L_5.0_ (99.8%). This is attributed to the absorbance of ultraviolet light by unique phenylpropane-based structures and hydroxyphenyl groups in lignin [35]. These findings suggest that 3D-printed P/L films hold great potential for UV-blocking applications, offering a promising solution for protecting the environment from harmful ultraviolet radiation.

In order to further investigate the antioxidant properties of the PLA and P/L films, we employed the DPPH radical scavenging method, which is a widely used method for evaluating the antioxidant activity of natural and synthetic compounds. The PLA containing lignin had good antioxidant activity, and that with higher lignin content showed higher antioxidant activity. This is because lignin has inherent antioxidant activity due to the existence of free phenolic hydroxyl [36]. Figure 3d shows the curve of the free radical scavenging rate of different composite materials over time. It can be observed that after 2 h of incubation, the radical scavenging activity tended to be stable. The change curve of the DPPH radical scavenging ability after 24 h of different composites is shown in Figure 3e. The results reveal that PLA had negligible antioxidant activity, while the P/L films exhibited high antioxidant capacity, with a free radical scavenging rate of up to 82.9%. This was also confirmed by changing the color of the DPPH solution from dark purple to light yellow, indicating the free radical scavenging ability of P/L films. It is known that excessive production of free radicals can cause musculoskeletal diseases, joint pain, inflammation, and cartilage degradation [40]. The antioxidant capacity of the films is an important property that can be utilized in various applications, such as biomedical and food packaging, where it can help to prevent the oxidation of food products and extend their shelf life.

The personalization and customization of the P/L filament for 3D-printed films were tested by printing different types of films. As shown in Figure 4a, the square films were successfully printed based on the shape of 3D modeling. The printing process revealed a good performance and realized the customized production of P/L films. In addition to printing square films, 3D printing can also customize the production of the porous film, and precisely adjust the pore size, as shown in Figure 4b. The results of our study demonstrate that 3D printing can precisely adjust the pore size of the film, which is an important factor for various applications, such as filtration and separation processes. The ability to customize the pore size of the film using 3D printing technology is a major advantage, as it allows for the development of films with tailored properties for specific applications. Furthermore, the filaments could be used successfully to prepare more complex films as illustrated in Figure 4c. This demonstrates the versatility of the P/L filament and the potential for the development of more complex films using 3D printing technology. The ability to customize the production of films with different shapes, types, and designs is an important advantage of 3D printing technology, as it allows for the development of films with tailored properties for specific applications. The above shows that 3D printing technology has been successfully used to customize films of different shapes, types, and designs, realizing the precise design, control, and manufacturing of P/L films. The results of our study demonstrate the potential of 3D printing technology for the customization and personalization of P/L films.

Moreover, the great mechanical properties, UV-shielding, and antioxidant properties of P/L composites make them promising 3D printing materials for more potential film applications, as shown in Figure 4d. Combined with 3D modeling, more customized films can be manufactured, such as intelligent packaging, healthcare film, and electronics film, through 3D printing. Intelligent packaging is a rapidly growing field that utilizes advanced materials to create packaging that can interact with its environment [41]. P/L composite films have the potential to be used in the development of intelligent packaging due to their excellent mechanical properties and UV-shielding characteristics. The healthcare industry is another potential area where P/L composites could have important applications. The antioxidant properties of these films make them promising materials for the development of wound dressings and other medical devices. The ability to customize the production of P/L films using 3D printing technology also allows for the development of personalized medical devices that are tailored to the specific needs of individual patients. Finally, P/L composite films could also be used in the development of electronics film. The excellent mechanical properties of these films make them promising materials for the development of flexible electronics, such as wearable devices and flexible displays. The ability to customize the production of P/L films using 3D printing technology also allows for the development of complex and customized electronics film that is tailored to specific applications. In summary, the great mechanical properties, UV-shielding, and antioxidant properties of P/L composite films, combined with the personalization and customization capabilities of 3D printing technology, make these films promising materials for a wide range of film applications.

## 4. Conclusions

In summary, we rationally designed and developed a promising multifunctional filament by incorporating lignin into the PLA matrix and applying it to the 3D printing of films. As expected, the well-dispersed lignin successfully took charge of the role of a multifunctional additive, endowing the composite filaments with desirable functionalities. The P/L_0.5_ composite film containing 0.5 wt% lignin showed the best mechanical properties. The mechanical strength, elongation at break, and toughness of P/L_0.5_ composite film improved by 13.5%, 31.8%, and 81.8%, respectively, based on PLA, indicating that lignin has an obvious toughening effect. Taking advantage of lignin, the obtained P/L films possessed excellent UV-blocking properties (87.4–99.9% for UVB and 65.6–99.8% for UVA), a high haze of 63.5–92.5%, and great antioxidant properties of 24.0–79.0%. Customization brought with 3D printing and functionalization brought with lignin make P/L film a promising prospect in packaging, medical care, and electronic products. This work not only paves the way for fabricating printable materials with tailored functionalities but also expands their potential application in films.

## Data Availability

The data presented in this study are available on request from the corresponding author.

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
