# Peer review of "3D-Printed Polylactic Acid/Lignin Films with Great Mechanical Properties and Tunable Functionalities towards Superior UV-Shielding, Haze, and Antioxidant Properties"

_polymers, 2023, doi:10.3390/polym15132806_

Round 1

Reviewer 1 Report

·         - Please confirm whether the ideas in Introduction are smooth and the transition is natural. The reasons for the use of various raw materials are clearly explained, but the story is not good enough.

·         - The authors should re-organize the introduction to increase the readability and highlight the innovation of the manuscript compared with other works.

The manuscript needs revision for language and grammar.

Reviewer 2 Report

Ye et al have 3D printed PLA-Lignin films by incorporating Lignin in PLA matrix. These films exhibit good mechanical, UV-shielding, haze and antioxidant properties compared to PLA.  These were also validated experimentally. The following comments need to be addressed.

1.      1. In page 1. Line no. 27-39, introduction section 2nd paragraph is same as 1st.

2.      The 3D printing parameters like layer thickness, print speed, etc should be mentioned in the manuscript.

3.      The details of DPPH methods are also missing. How the test was carried? which solvent was used, etc.

4.      Analysis like FTIR, UV-vis, TGA should also be carried out on 3D printed PLA-lignin’s films along with filaments to support the result.

5.      In figure 3 caption is incomplete. Captions for (b) and (c) are missing and in (d) data with time for free radical scavenging (RSA) is missing and only 24 hour data is presented.

No comments 

Reviewer 3 Report

Ye et al. presented the fabrication of a high-performance polylactic acid (PLA)/lignin film using fused deposition modeling 3D printing. The 3D-printed PLA/lignin films, as compared to pure PLA material, exhibited improved mechanical properties. In addition to that, the 3D-printed composite also showed superior UV-shielding, haze, and antioxidative capabilities. The work presented here is interesting, with comprehensive experiments well performed to support the authors’ major claims.

However, the introduction section could better highlight the concepts of the work, as some arguments seem unfounded. To strengthen the selling points, some revisions are necessary. (1) While films and 3D printing are indeed critical to the development of manufacturing, 3D printing is predominantly recognized for creating ‘3D’ structures, not 2D films. It would be helpful if the authors could more clearly articulate how 3D printing improves the accuracy of size, shape, and thickness in film production. (2) The claim that 3D printing is ‘complex or requires expensive and non-degradable components’ could use more nuance. Modern 3D printing includes a wide range of technologies, some of which are relatively simple and use affordable, eco-friendly materials. (3) As the authors acknowledged, the integration of lignin with PLA and its benefits are well-documented. If the authors' work is merely combining these two well-studied facts, it would be beneficial to more clearly delineate what new insights or contributions this work provides over existing studies. Overall, this manuscript will be suitable for publication after the previous concerns are addressed. The following are some technical comments.

(1)   The term ‘antioxidant’ refers to substances that inhibit oxidation reactions. From the context, the authors are referring to the antioxidative properties of the film. It would be necessary to double-check and revise if needed.

(2)   Abbreviations need to be defined in the main text of the manuscript when the term shows up for the first time. A few abbreviations are not defined, for example, P/L was not defined in the abstract, DTG was not defined in the manuscript, RSA was only defined in the figure caption, and DPPH was not defined when this term appeared for the first time.

(3)   The description of the preparation of P/L filament does not provide sufficient information. Key information such as the particle size of PLA pellets and lignin powders is missing. Moreover, it would also be helpful to provide the mixing speed of the mixer and the extrusion rate for the fabrication of the filament. Additionally, the mixing temperature, extrusion temperature, and 3D-printing temperature are chosen to be 170, 150, and 200 C respectively, and it would be much appreciated if the authors can comment on why these temperatures were selected for a specific process.

Please see the previous comments for English improvement. Thanks.

Round 2

Reviewer 2 Report

Authors have addressed all the concerns. I wish them good wishes